# Defining Preferred and Natural Robot Motions in Immersive Telepresence from a First-Person Perspective

### Katherine J. Mimnaugh
katherine.mimnaugh@oulu.fi
University of Oulu
Oulu, Finland

### Markku Suomalainen
University of Oulu
Oulu, Finland
markku.suomalainen@oulu.fi

### Israel Becerra
Centro de Investigación en
Matemáticas
Guanajuato, México

### Eliezer Lozano
Centro de Investigación en
Matemáticas
Guanajuato, México

### Rafael Murrieta-Cid
Centro de Investigación en
Matemáticas
Guanajuato, México

### Steven M. LaValle
University of Oulu
Oulu, Finland

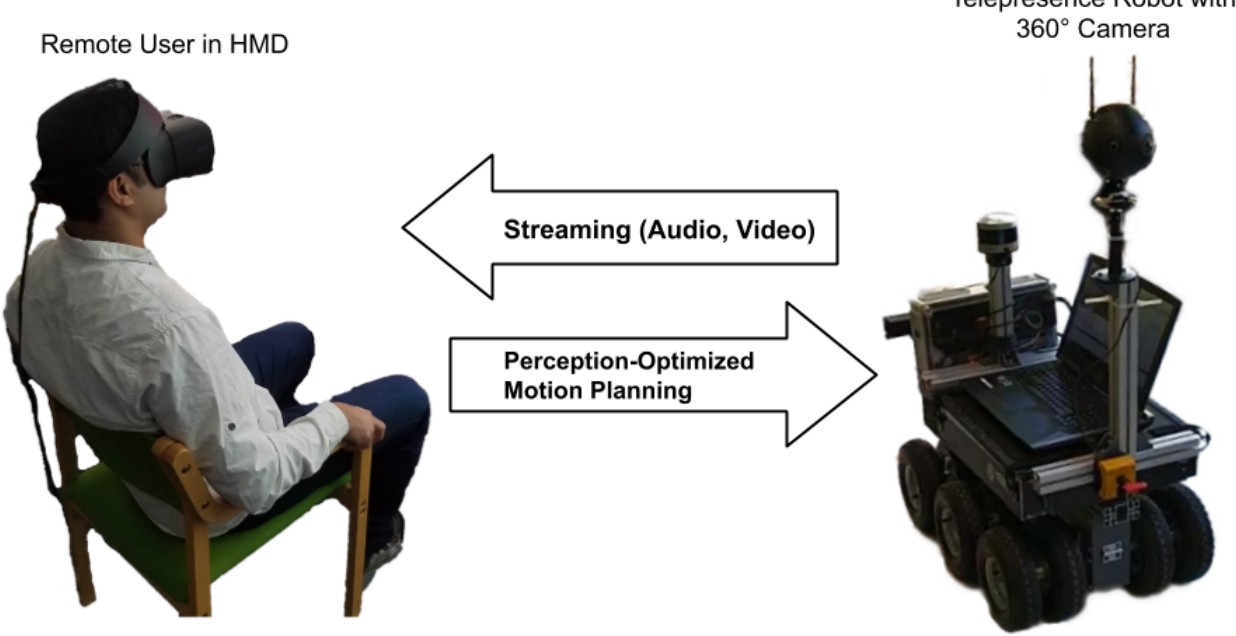

**Figure 1: An overview of immersive robotic telepresence through an HMD; the remote user can be telepresent in a remote location and feel as if they are really there.**

## ABSTRACT

This paper presents some early work and future plans regarding how the autonomous motions of a telepresence robot affect a person embodied in the robot through a head-mounted display. We consider the preferences, comfort, and the perceived naturalness of aspects of piecewise linear paths compared to the same aspects on a smooth path. In a user study, thirty-six subjects (eighteen females) watched panoramic videos of three different paths through a simulated museum in virtual reality and responded to questionnaires regarding each path. We found that comfort had a strong effect on path preference, and that the subjective feeling of naturalness

also had a strong effect on path preference, even though people consider different things as natural. We describe a categorization of the responses regarding the naturalness of the robot's motion and provide a recommendation on how this can be applied more broadly. Although immersive robotic telepresence is increasingly being used for remote education, clinical care, and to assist people with disabilities or mobility complications, the full potential of this technology is limited by issues related to user experience. Our work addresses these shortcomings and will enable the future personalization of telepresence experiences for the improvement of overall remote communication and the enhancement of the feeling of presence in a remote location.

## CCS CONCEPTS

• **Human-centered computing → Virtual reality**; **User studies**; **Empirical studies in HCI**; • **Computer systems organization → Robotic autonomy**; **Robotic control**;

## KEYWORDS

Human-Robot Interaction, Adaptive Robotic Behavior, Telepresence, Virtual Reality, Robotics, Motion Planning, User Studies

**ACM Reference Format:**
Katherine J. Mimnaugh, Markku Suomalainen, Israel Becerra, Eliezer Lozano, Rafael Murrieta-Cid, and Steven M. LaValle. 2021. Defining Preferred and Natural Robot Motions in Immersive Telepresence from a First-Person Perspective. In *Proceedings of VAM-HRI '21: ACM/IEEE International Workshop on Virtual, Augmented, and Mixed-Reality for Human-Robot Interactions (VAM-HRI '21).* ACM, New York, NY, USA, 4 pages.

## 1 BACKGROUND

Currently, the most scalable technology with the potential to achieve the feeling of being immersed in a remote location is a panoramic camera streaming to a Virtual Reality (VR) Head-Mounted Display (HMD); see Fig. 1. The immersion provided by an HMD can facilitate more natural interaction and thus ease the difference between telepresence and physical presence [10]. Unfortunately, there are several concerns regarding the use of telepresence with an HMD that do not typically occur in telepresence with a Fixed Naked-eye Display (FND, a traditional display such as on a computer or a phone). The velocities that the robot uses or the passing distances to objects and walls, which would otherwise be comfortable in FND telepresence, can be uncomfortable in immersive telepresence. Second, a serious consequence of using an HMD is *VR sickness* [6], which can result in the user experiencing nausea and vertigo.

Although there are known methods to reduce VR sickness [2], such as reducing the Field-of-View (FOV) [11], these techniques may have a negative effect on the feeling of being present [13]. However, by planning autonomous trajectories for the robot while avoiding motions known for contributing to VR sickness, the feeling of sickness can be decreased without sacrificing the feeling of presence. Additionally, speeds and distances to objects can also be taken into account to make the experience comfortable for the user. Thus, to effectively adapt the robot's motion planning criteria priorities to each user, the impacts of these components on the user must be analyzed. For example, turns have been shown to play a role in VR sickness, [3], but there is evidence that the duration of

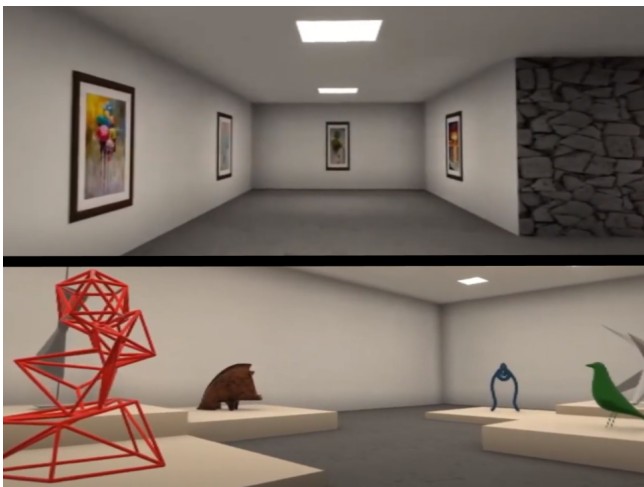

**Figure 2: A screenshot inside the virtual museum, seen by participants from the point of view of the virtual telepresence robot. The entrance hallways is seen on top, and the main gallery in the museum is shown on the bottom.**

the turn is more important than the total angle of rotation [14]. This knowledge was used to plan piecewise linear trajectories for a robot, with rotations in place, which resulted in reduced VR sickness in an immersive telepresence experience [1].

However, there are other aspects to consider besides reducing VR sickness; issues such as the path not feeling natural or qualities of the turns when rotating in place may detract from the experience. These issues are typically not a problem in telepresence through an FND, since it is the immersion experienced through the near-eye displays of an HMD that allow the user to really feel them. These issues are also not frequently studied in pure VR research, since motions in VR are usually either directly controlled by the user, or teleportation is used. Although these methods for locomotion in VR may be more comfortable for the user, teleportation is infeasible in telepresence and manual control is perceived as bothersome in large environments [9]. Thus, there is a need to research autonomous motions to overcome these obstacles in immersive telepresence.

Negative impacts, like VR sickness, are not the only concern when designing autonomous telepresence robot trajectories. An interesting, yet unexplored, positive element of the telepresence experience is the *naturalness* of the robot's motions. Some work has discussed attempts to make robot motion more natural [5] or deployed voiced commands to make robots perform more naturally [8]. However, to the authors' knowledge, there is no widely accepted definition of natural motions for mobile robots. Moreover, from the perspective of the user embodying the telepresence robot, there may not currently be any research on whether people even prefer motions that they would perceive as natural, or more generally what kind of trajectories are preferred for an autonomously moving telepresence robot to take.

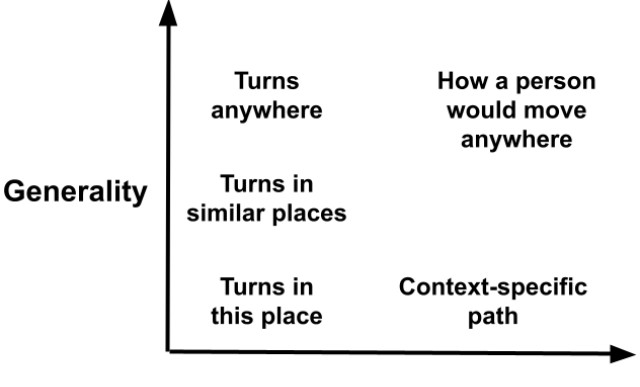

**Figure 3: Conceptualization of the dimensions on which subjects consider movement to be natural. Horizontally is the time frame dimension. Things closer to the origin are more immediate (turns), whereas away from the origin, longer term, relates to things like the path. On the generality dimension, things closer to the origin are specific to a place (in this museum) whereas things away from the origin are more general (in any setting). The units of each dimension are not discrete and can vary by individual.**

## 2 METHODS

We performed an experiment where the subjects wore an HMD and watched three pre-recorded trajectories of a virtual telepresence robot moving through a virtual museum. A screenshot from inside the museum can be seen in Fig. 2, and the videos (in 2D format) can be viewed at: https://youtu.be/FPbrIlw3OhY. The data was collected in February 2020, before coronavirus restrictions were enacted in Finland. Two of the trajectories were piecewise linear trajectories with 45° or 90° turns, optimized for the least amount of turns and for the shortest path (see [1] for the path planning), while the third path was a smooth path created with a Rapidly-exploring Random Trees (RRT) algorithm [7]. We selected the RRT because it is commonly used in robot motion planning, it respects the differential drive robot system's dynamics while planning the trajectory in 5-dimensional space, and because there is evidence that RRT paths are considered to be human-like [12]. We asked the subjects to rate aspects of the paths on a Likert-scale, and to select their preferred path, the most comfortable path, and the most natural path. We also asked them open-ended questions regarding why they made those selections.

The environments were created with the Unity 3D game engine and displayed to subjects as 360 degree videos played through the Virtual Desktop application into an Oculus Rift S. Subjects were seated in a stationary chair and able to look around while the videos were played. Thirty-six subjects from campus and the local community were run in a gender and path counterbalanced within-subjects design, with each subject viewing all three of the stimulus videos and completing a Simulator Sickness Questionnaire (SSQ, [4]) and questionnaires regarding their experience on each path. Subjects signed a consent form before participating and were debriefed after the study was completed.

## 3 RESULTS

We set out to examine the interplay between the preference, comfort, and perceived naturalness of robot motions for first-person experiences in VR telepresence. We found the most salient aspect regarding comfort was in relation to the turns. Comfort had a strong association with preference, but not with naturalness. We also found that, after comfort, preference for a particular path was influenced the most by forward speed and the turns. Preference was strongly associated with the users' perceived naturalness, which was primarily determined by the ability to see salient objects, the distance to the walls and objects, as well as the turns. Participants favored the paths that had a one meter per second forward speed and rated the path with the least amount of turns as the most comfortable. The piecewise linear paths with rotation in place were preferred and selected as the most natural more often than the smooth path where rotation and translation occurred simultaneously. Finally, we also found that the robot's speed and passing distances had a significant impact on the users, and must be carefully considered when implementing immersive telepresence.

Naturalness is a topic often mentioned as a goal in robotics publications, but there is no widely accepted definition of naturalness and it can mean different things to different people. In order to further elucidate how naturalness is thought of by people interacting with a robot, we asked participants open-ended questions regarding the naturalness of the robot's motions. We examined the answers on why people considered a certain path to be the most natural and found that there seem to be two dimensions (see Fig. 3) on which people interpreted the given definition of naturalness, "how you would have moved through the museum if you were in control": time frame, and generality. On the time frame dimension, responses regarding either how the robot performed the turns (fifteen comments, twenty-five percent of the total comments on naturalness) or how close to walls and objects the robot moved (thirteen comments, twenty-two percent) can be considered to reflect something more immediate, while comments related to things further in time referred to aspects of the whole path through the museum (fifteen comments, twenty-five percent). Regarding the generality dimension that describes either context-specific or more general behavior, subjects specifically mentioned the museum setting in seven answers spread across all paths, but some comments regarding turning, for example, could be considered as not specific to the context.

The two dimensions in Fig. 3 depict a categorization of certain aspects of the answers regarding naturalness when subjects were queried on how they would have moved through the museum if they were in control. Answers about naturalness seemed to generally reflect these two dimensions, though how naturalness is conceptualized will include significant individual variation. Furthermore, it should be noted that these answers are a expression of what participants though of as natural in a particular context, the virtual museum. Whereas some answers were specific to this setting (*"I would visit one side first and then go to the other side of the museum, slowly exploring. So the first one was more natural"*), others

discussed naturalness more generally (*"the walking speed was closet [sic] to my own"*). Although the situation in which these answers were generated was quite distinct, the authors believe that people think of naturalness on these two dimensions in many other cases and situations, which our subsequent work will examine.

## 4   BROADER IMPACT

When attempting to make a robot move "naturally," researchers should take individual variation into account and consider the two dimensions on which it might be perceived. For example, researchers can consider that natural motion in a certain context might not be natural in other contexts, which we found in answers where subjects contrasted how they would move in a museum (*"I think in real life I would move similarly to the 3rd path, because it would allow me to view the items better (closer). Also, it simulates the hesitation of a visitor when going in around a museum"*) versus how they consider themselves to move more generally (*"Walking through a corridor in straight line felt more natural than it was in the first path where the movement was from one corner towards another"*). Researchers can also consider that the naturalness of parts of the path like the speed and how the turns are executed (*"This speed is more natural for a scenario or location such as this. However, the turns were a bit sharp and you feel like you're too close to the objects"*), or the whole path that is taken (*"It felt most like a path that an actual human would take, getting closer to inspect the paintings"*), can each be considered to exhibit naturalness or a lack thereof. All of the individual pieces of a trajectory may be construed as natural or not natural in conflicting ways, so there does not seem to be one universal method to categorize robot motion as natural or not. Therefore, care should be exercised in the third-party classification of the naturalness of robot motion; indeed what a roboticist designing the motion or a reviewer reading the paper might assess as natural may be quite distinct from what a general member of the public may appraise as being natural. Thus the best determination of naturalness is, therefore, attained through the evaluation of research participants.

## 5   FUTURE WORK

Future work will will test these same trajectories through the museum, except without any artwork present. Since many people regarded path qualities to be related to their ability to see objects in the museum, the artwork had a significant impact on the subjects' responses regarding preference, comfort, and naturalness. In contrast to the first study with the art, we hypothesize that the turns will still have the largest impact on comfort, and that the path with the least amount of turns will be the preferred path when moving through an empty room. We also postulate that the least turns path will be rated as the most natural because it is piecewise linear. The resulting comparisons will further elucidate how naturalness is conceptualized more broadly and will provide further clarification on user preferences from a first person perspective regarding turn speed, forward speed, trajectories, and comfort for the physical movements of immersive telepresence robots.

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
