# OpenReview forum: "Defining Preferred and Natural Robot Motions in Immersive Telepresence from a First-Person Perspective"
_humanrobotinteraction.org/HRI/2021I/Workshop/VAM-HRI — VAM-HRI 2021 Oral_

### Official Review · AnonReviewer1 · 2021-03-03
**Review: Preference of Trajectories in Immersive/VR Telepresence**

**Rating:** 8
**Confidence:** 4

**Review:**

This paper presents preliminary results from a human subjects study. Participants observed 3 different predetermined trajectories via a virtual reality head-mounted display, providing them telepresence in a museum. Two of the trajectories followed piecewise linear paths and the other used RRT. It is mentioned that RRT was chosen because it creates human-like paths, however participants preferred the other, piecewise linear paths instead. Why might this be? Is this also an area for future work? It is also unclear which trajectories had varied speeds - the Methods section describes the 3 trajectories that subjects saw, however only in the Results section is speed referenced. In what cases was speed varied? There is also significant discussion of naturalness - a useful theme to cover, which hopefully resurfaces in the workshop. This paper is well done, highlights relevant results, and I recommend accept.

---

### Decision · Program_Chairs · 2021-03-06

Accept (Oral)